# A Pilot Randomised Controlled Trial Evaluating a Regenerative Epithelial Suspension for Medium-Size Partial-Thickness Burns in Children: The BRACS Trial

Anjana Bairagi [1,2,3,*], Zephanie Tyack [1,3,4], Roy Kimble [1,2,4], Dimitrios Vagenas [5], Steven M. McPhail [3,6] and Bronwyn Griffin [1,7]

1  Centre for Children's Burns and Trauma Research and Pegg Leditschke Children's Burns Centre, Queensland Children's Hospital, Brisbane, QLD 4101, Australia
2  Burns Trauma Research, Centre for Children's Health Research, Queensland University of Technology, Brisbane, QLD 4101, Australia
3  Australian Centre for Health Services Innovation (AusHSI), Centre for Healthcare Transformation, Queensland University of Technology, Brisbane, QLD 4059, Australia
4  Child Health Research Centre, Faculty of Medicine, The University of Queensland, Brisbane, QLD 4101, Australia
5  Research Methods Group, Faculty of Health, Queensland University of Technology, Brisbane, QLD 4059, Australia
6  Digital Health and Informatics, Metro South Health, Brisbane, QLD 4102, Australia
7  NHMRC Centre of Research Excellence Wiser Wound Care, Griffith University, Brisbane, QLD 4222, Australia
*  Correspondence: a.bairagi@qut.edu.au

**Abstract:** Background: There is little evidence regarding the efficacy of Regenerative Epidermal Suspension (RES™) management for paediatric partial-thickness burns. The Biobrane® RECELL® Autologous skin Cell suspension and Silver dressings (BRACS) Trial evaluated three dressings for the re-epithelialisation of partial-thickness burns in children. Methods: Eligible children (age ≤ 16 years; ≥5% TBSA; ≤48 h of injury) were randomised to silver dressings, RES™/Biobrane® or Biobrane®. The measured outcomes were the time to re-epithelialisation (primary outcome), pain, itch, intervention fidelity, treatment satisfaction, health-related quality of life, health resource utilisation and adverse effects. Results: The median time to re-epithelialisation in days was no different for RES™/Biobrane® at 12 (IQR: 5.6–18.4; $n = 7$) and slower by two days for Biobrane® at 14 (IQR: 6.3–21.7; $n = 7$) when compared to silver dressings 12 (IQR: 3.7–20.3; $n = 8$). Reduced pain, fewer infections, no sepsis, no skin graft, and the lowest impact on health-related quality of life were reported in the RES™/Biobrane® group compared to other groups. Due to the COVID-19 pandemic, recruitment suspension resulted in a smaller cohort than expected and an underpowered study. Conclusions: The pilot trial findings should be interpreted cautiously; however, they indicate that a fully powered randomised controlled trial is warranted to substantiate the role of RES™ for medium to large paediatric partial-thickness burn management.

**Keywords:** child; burn; re-epithelialisation; randomised controlled trial; autologous skin cell suspension; RECELL

## 1. Introduction

Despite advancements in burn care, scald injuries are the most commonly managed burn injury for children in Australia and worldwide [1–3]. These injuries can vary in thickness, presenting with a combination of superficial partial, mid-dermal, or deep partial-thickness burn depths and in re-epithelialisation time. Predictors of delayed re-epithelialisation include the burn depth, total body surface area burned (TBSA-B), injury mechanism, age of burn at presentation to burn centre and pain scores [4]. The risk of scar hypertrophy in children was reported to increase with delayed re-epithelialisation [5–8]. In

cases where the TBSA-B > 5%, management challenges include mitigating the potentially severe morbidity associated with the injuries (e.g., sepsis, severe pain and distress, scar contracture) and the increasing demands for health care resource utilisation at treating centres. In the context of large TBSA-B burns across Australia and New Zealand, less than three percent of children sustain burns greater than 20% TBSA-B [9]. Hence, modern burn wound care is focused on expediting wound re-epithelialisation in which wound closure can be achieved spontaneously using dressings, surgically (by skin graft) or both to optimise outcomes following injury.

Silver-impregnated dressings, such as Acticoat® (Smith and Nephew, Hull, UK) with Mepitel® (Mölnlycke, Göteborg, Sweden) or Mepilex Ag® (Mölnlycke, Göteborg, Sweden), were standard care at the study site during this trial [10]. The reported advantages of silver dressings for paediatric burn wounds include faster re-epithelialisation, cost-effectiveness, less infections, and reduced pain [10–13]. Non-cultured autologous skin cell suspensions (ASCS) have been used as biological dressings to treat burn wounds for over three decades [14,15]. However, few studies have evaluated the efficacy of ASCS such as the Regenerative Epidermal Suspension (RES™), which is prepared with the RECELL® autologous cell harvesting device (AVITA Medical, Valencia, CA, USA), for paediatric burn wound re-epithelialisation. Biosynthetic, bilaminar, silicone-based skin substitutes such as Biobrane® (Smith and Nephew, Hull, UK), have been utilised in paediatric burn wound care with reported shortened re-epithelialisation times, reduced pain, and varied infection rates [12,16–20].

The only study evaluating these three wound-management options in paediatric burn wounds was a pilot randomised trial of 13 children in a single burn centre in Australia where burn wounds treated with RES™/Biobrane®. Biobrane® alone required less than half the time to re-epithelialisation (TTRE) when compared to silver dressings [17]. There were no wound infections nor sepsis in the active control group (silver dressings). However, there was a single wound infection in the RES™/Biobrane® and Biobrane® alone groups and a single case of sepsis in the Biobrane® alone group [17].

Of the several commonly available dressings for treating medium- to large-size partial-thickness burns, three types were evaluated in the BRACS Trial: silver- (nanocrystalline impregnated silver), biological- (non-cultured ASCS) and silicone- (biosynthetic skin substitute) based dressings. Burn wound re-epithelialisation in days was assessed as the primary outcome. Secondary outcomes evaluated at the primary endpoint of >95% TTRE included pain, itch, ease of dressing application, intervention fidelity, treatment satisfaction, scar severity, health related quality of life, health resource utilisation and adverse effects.

## 2. Materials and Methods

This study was designed as a parallel-group, single-centre, randomised controlled trial (RCT) [14]. However, due to the indefinite cessation of recruitment in response to the COVID-19 pandemic, a smaller-than-anticipated sample resulted, and the trial was underpowered for definitive findings. It is thus presented as a pilot RCT using the CONSORT guidelines [21]. The study site is a specialist paediatric hospital at which >1200 new patients are treated for acute burn injuries annually.

Children presenting to the study site who met the inclusion criteria (≤16 years with acute burn injury, <48 hours after the injury, burns of superficial to mid-dermal partial-thickness burn depth and ≥5% total body surface area burned) were recruited. Informed consent was obtained from all subjects involved in the study. Once enrolled, baseline demographic data (included participant ethnicity, language, Fitzpatrick skin type and co-morbidities) were collected. Eligible participants were randomised to one of three interventions: Silver dressings (Acticoat® with Mepitel® or Mepilex Ag®); RES™/Biobrane® or Biobrane®.

After randomisation, the initial dressing for the assigned intervention was applied under a general anaesthetic for all participants. Peri-operative, intravenous, antimicrobial prophylaxis was administered at the first dressing application, followed by the non-excisional

debridement of the burn wounds with a soap-free wash and sterile water. At this point, the previously determined TBSA-B was assessed by the attending burn surgeon in addition to the NSW ITIM [22,23] and E-Burn [24,25] mobile burn size measurement applications. The E-Burn application was thought to be easier to use; however, it was not previously validated when compared to the NSW ITIM mobile application for the assessment of a burn wound area. A recent study assessing large paediatric burn injuries compared E-Burn to the Mersey Burns application and the Lund and Browder chart and found that the E-Burn application was the easiest to use [25]. Similarly, the presenting burn depth was assessed by the attending burn surgeon and objectively measured with laser doppler imaging [26–28]. The application of the silver dressings, RES™ and Biobrane® were completed as detailed in the study protocol [14].

Following the initial dressing application, dressing changes thereafter were conducted every three to five days until ≥95% re-epithelialisation, as per standard clinical practice at the participating centre. The timing of the subsequent dressing changes was determined by the attending burn surgeon and was completed under anaesthesia when clinically indicated. At dressing changes, silver dressings were replaced completely. For the RES™/Biobrane® and Biobrane® groups, the lifted edges of Biobrane® were trimmed and secondary dressings changed until all the Biobrane® had lifted completely off the wound. Data collection was completed at the initial dressing application (baseline) following randomisation and then at each dressing change until the primary endpoint of ≥95% burn wound re-epithelialisation. Data collection then continued until the final assessment at 12-months post burn injury.

The primary outcome was the time to burn wound re-epithelialisation, determined by the burn surgeon and defined as the number of days to ≥95% spontaneous wound re-epithelialisation post burn injury. The secondary TTRE outcome was a masked review of burn wound images by a panel of burn clinicians and two-dimensional (2D) area assessment as described in the protocol [14].

The secondary outcomes measured were pain, itch, ease of dressing application, intervention fidelity, treatment satisfaction, scar severity, health related quality of life, health resource utilisation and adverse effects. Pain was reported by clinicians using the Face, Legs, Activity, Cry, and Consolability Pain Scale (FLACC) [29] as a behavioural observation measure. Parent/guardians proxy-reported pain and itch intensity with an 11-point numeric rating scale (NRS-P Proxy and NRS-I Proxy, 0 = no pain or itch, 10 = worst imaginable pain or itch) [30]. Children greater than eight years self-reported pain and itch intensity with the revised Faces Pain Scale—Revised (FPS-R) [31] and an 11-point numeric rating scale (NRS-P and NRS-I, 0 = no pain or itch, 10 = worst imaginable pain or itch). The ease of dressing application (EDA) was assessed at each dressing application using a study-specific questionnaire that consisted of three questions regarding application ease, conformability, and duration as well as a free text section for comments. Wound intervention fidelity was assessed with a pre-specified checklist for each intervention group. Both clinicians and parents/guardians rated treatment satisfaction with an 11-point numeric scale (NRS-TS, 0 = not satisfied, 10 = extremely satisfied).

Burn scar relocation was completed at each 3-, 6- and 12-month follow-up visit. All scar severity data were collected with reference to the most severe site of the scar as identified by the parent/guardian of the participant at the follow-up visit. The scar thickness was obtained using the Venue40 MSK® Ultrasound machine (GE Healthcare, Fairfield CT, USA) by an investigator aligned with the study. An experienced sonographer, masked to the assigned intervention group, measured the average of three measurements for each scar. Similarly, scar colorimetry was collected with the DSM II ColorMeter® (Cortex Technology, Hadsund, Denmark) using the CIE Lab colour space system (L*, a*, b*) [32, 33], as documented in the protocol [14]. The scar severity was rated by clinicians and parents/guardians with the Patient Observer Scar Assessment Scale (POSAS) [34]. The health-related quality of life (HRQoL) was rated by all the participants/guardians using the Brisbane Burn Scar Impact Profile (BBSIP) [35–41] and the nine-item Child Health Utility 9D (CHU9D) [42–46]. The CHU-9D responses obtained from a parent/guardian of each

participant were transformed to a multi-attribute utility score on a scale of zero to one in which the higher the score, the higher the HRQoL. Health resource utilisation data collected for each participant included intervention and hospitalisations that combined setting and labour time, among other cost buckets.

Recorded adverse events included wound infection, allergic reaction, sepsis, unplanned admission to the intensive care unit, burn progression, need for a split-thickness skin graft and others. The adverse events were graded using the Clavein–Dindo classification for surgical complications [47,48]. A regular report was submitted to the Safety Monitoring Group, which comprised two independent burn surgeons and one clinical nurse consultant who was not aligned with the study and was from a different burn centre.

The initial sample size estimate was calculated based on the ability to detect a minimal, clinically important difference of four days for re-epithelialisation with a total sample of 84 participants (28 per group). The randomisation sequence was prepared by a biostatistician and entered into REDCap by an independent third party not affiliated with the study. Randomisation and allocation ((1:1:1 ratio) with block size of eight) were assigned electronically using the Research Data Capture (REDCap) randomisation module [49]. As much as possible, participants and parents/guardians were masked to their allocations. The panel of burn clinicians evaluating the burn wound re-epithelialisation were also masked to the assigned intervention.

*Statistical Methods*

The baseline demographic data and secondary outcomes were described with summary statistics. Medians and interquartile ranges (IQR) were used for non-parametrically distributed continuous data [50]. Means and standard deviations (SD) were used for normally distributed continuous data. Numbers and percentages were used for categorical outcomes [50]. The inter-rater reliability of TBSA-B calculation by the burn surgeon compared to the mobile applications (E-Burn, NSW ITIM) was assessed using intraclass correlations for agreement ($ICC_{agreement}$) with 95% confidence intervals (CI). A two-way random effects model for single measures was used [50]. The minimum standard for reliability was an $ICC_{agreement}$ of 0.70 for research purposes [51].

The primary outcome was analysed with Kaplan–Meier survival analysis, using a log-rank test with the TTRE as the main outcome and intervention group as the explanatory variable. The primary analysis sought to incorporate participants who underwent split-thickness skin grafting. As the time of grafting is often influenced by factors unrelated to the wound, such as operating room availability, a previously described method, which was employed to estimate the spontaneous time to re-epithelialisation using a dummy value, was sought. Six experienced paediatric burn surgeons (+/− nurses) with a collective >100 years of experience were individually surveyed on their predicted TTRE for the participants without an split-thickness skin graft; the conservative agreement was 28 days. This approach was taken before in published research [52,53]. The masked review of burn wound re-epithelialisation was analysed with a two-tailed Pearson correlation between the burn surgeon and a panel of experienced burn clinicians (three burn surgeons and one burn nurse) for each group.

The previously proposed quantitative analysis was not appropriate for all secondary outcome data due to the reduced number of participants. Consequently, the data were reported with descriptive statistics alone, using the median and interquartile range. For the same reason, the interim analysis at an enrolment of 30 participants was not completed. The responses recorded for the free text comment section for the ease of dressing application outcome were evaluated with a qualitative content analysis, utilising a deductive approach [54–56]. The responses were first divided according to intervention group (silver dressings, RES™/Biobrane® and Biobrane®) and then subdivided into two groups based on important time points at which the data were collected (baseline and dressing change) to allow for a meaningful interpretation of the data [57]. Due to the complex nature of the data, both *manifest* (broad surface structure) and *latent* (deep structure) analyses were used

where applicable. In addition, some responses were coded to more than one category where applicable. The calculation of the RECELL® ACHD (Autologous Cell Harvesting device) unit cost was based on the RECELL® 320 ACHD price as the RECELL® 1920 ACHD used in this trial had not been officially launched in Australia. An appropriate adjustment was made to accommodate for the difference in amount of RES™ prepared by the RECELL® 320 ACHD when compared with the RECELL® 1920 ACHD [58]. Due to the small sample size, the adverse event data were reported with descriptive statistics in lieu of a post-hoc inferential statistical analysis.

The data were analysed using an "Intention to Treat" approach. The statistical significance was set at $p < 0.05$ for the outcome of TTRE. The data set was analysed using IBM SPSS Statistics 26 (IBM Corporation, Armonk, NY, USA) software.

## 3. Results

### 3.1. Participant Recruitment

Recruitment was conducted over a 22-month period from 5 May 2018 to 30 March 2020. The recruitment was suspended indefinitely on 30 March 2020 as the study centre resources were diverted to manage the anticipated COVID-19 pandemic. A temporary restriction of aerosolised interventions in line with the organisational COVID-19 response requirements was implemented in favour of droplet application on 30 March 2020. Additionally, the original end date was changed from 30 November 2020 to 31 January 2021 for the completion of the final 12-month follow-up, as shown in Figure 1.

The first participant was enrolled on 4 June 2018, within one month of the commencement of recruitment. Of the 1669 children who presented to the study site with a new burn injury during the recruitment period, 110 children were screened for eligibility. Children were excluded ($n = 88$) from the study for reasons of not meeting criteria ($n = 56$), declining participation ($n = 4$), and other reasons (logistical reasons unrelated to the trial intervention) ($n = 28$). Thus, 22 children were enrolled in the study and were allocated to silver dressings ($n = 8$), RES™/Biobrane® ($n = 7$) or Biobrane® ($n = 7$). All participants received the assigned intervention and there was no loss to follow-up during dressing changes. Nineteen participants (86%) achieved ≥95% spontaneous burn wound re-epithelialisation. Three participants required a split-thickness skin graft (silver dressings: $n = 2$; Biobrane®: $n = 1$), Figure 1.

At the 12-month follow-up visit, 18 participants (82%) completed the final review. The 18% attrition rate comprised two participants lost to the follow-up at the three month review (assigned to Biobrane®) due to social reasons and two participants at the 12-month review (one assigned silver dressings and the other assigned RES™/Biobrane®) due to COVID-19 travel restrictions, as shown in Figure 1. Ten participants required scar management within 12-months post burn injury: silver dressings ($n = 5$), RES™/Biobrane® ($n = 1$) and Biobrane® ($n = 4$). Participants in the silver dressing and Biobrane® groups required one or more of the modalities for scar management, including topical silicone, a pressure garment, scar reconstruction, medical needling, and laser ablation. Only one participant in the RES™/Biobrane® group received scar management (topical silicone for a duration of six months) before the completion of scar management.

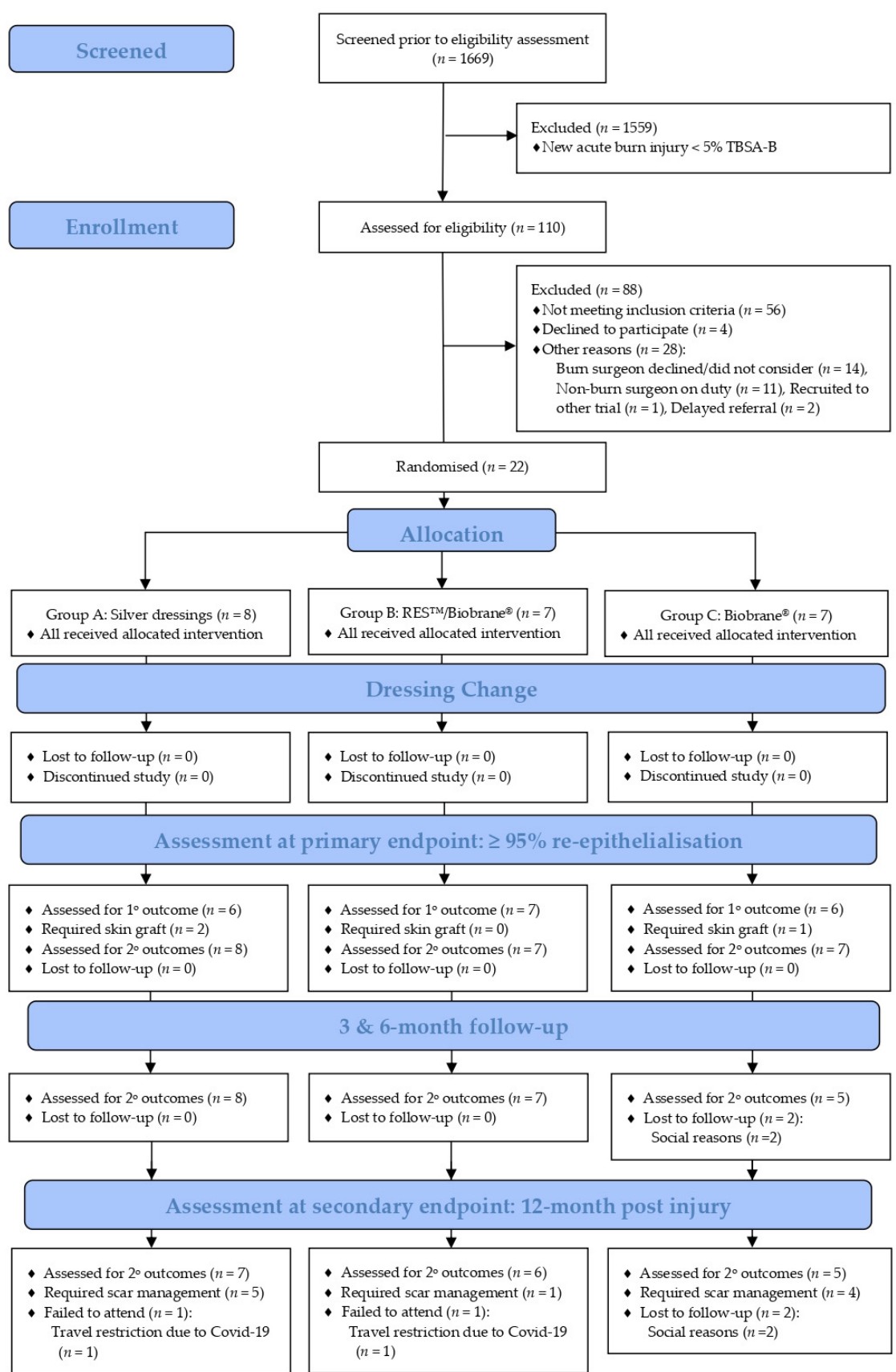

**Figure 1.** BRACS Trial CONSORT flow diagram. RES™: Regenerative Epidermal Suspension; TBSA-B: total body surface area burned; 1°: primary; and 2°: secondary.

### 3.2. Demographic Data

Twenty-two children were enrolled in the study with median age of 2.00 (IQR: 1.00–3.00) years. Of the children enrolled, 55% percent were female (*n* = 12). The participants were predominantly of European ethnicity (44%), Fitzpatrick skin type IV (44%) or V (44%) and had an English-speaking background (100%). The RES™/Biobrane® group had the least number of co-morbidities (*n* = 3) when compared to the silver dressing (*n* = 7), and Biobrane® (*n* = 6) groups, respectively, Table 1. The median TBSA-B was 10.0% (IQR: 6.00–15.25%), with an excellent intra-class correlation (ICC) of burn size calculated by the burn surgeon compared with both mobile applications (E-Burn [24], NSW ITIM [22,23]) for silver dressings (ICC $_{(2,1)}$ 0.921(95% CI 0.763, 0.982); $p < 0.0001$), RES™ /Biobrane® (ICC $_{(2,1)}$ 0.989(95% CI 0.958, 0.998); $p < 0.0001$) and Biobrane® (ICC $_{(2,1)}$ 0.982(95% CI 0.935, 0.999); $p < 0.0001$).

**Table 1.** Baseline participant characteristics.

| Parameter (*n*) | Silver Dressings ‡ (8) | RES™/Biobrane® (7) | Biobrane® (7) |
|---|---|---|---|
| **Age (years) *** | 1.50 (1.00–2.75) | 1.00 (1.00–2.00) | 2.00 (2.00–4.00) |
| Age (years, Min–Max) | 0.00–14.00 | 1.00–14.00 | 1.00–9.00 |
| **Gender (*n*)** | M4: F4 | M5: F2 | M1: F6 |
| **Ethnicity (*n*)** | | | |
| European | 3 | 2 | 3 |
| Aboriginal or Torres Strait Islander | 1 | 1 | 2 |
| Asian | 2 | 2 | 1 |
| Middle Eastern | - | - | - |
| African | - | 1 | - |
| Other | 2 | 1 | 1 |
| **Fitzpatrick Skin Type (*n*)** | | | |
| I | 1 | - | - |
| II | - | 1 | - |
| III | 2 | 2 | - |
| IV | 1 | 4 | 3 |
| V | 4 | - | 4 |
| VI | - | - | - |
| **Co-morbidities (*n*)** | | | |
| Allergies | 1 | - | - |
| Skin disorders | 2 | - | 1 |
| Physical disability | - | - | - |
| Learning disability | 1 | 1 | 1 |
| Mental health illness | - | 1 | 1 |
| PTSS | 1 | - | - |
| Visual Impairment | - | - | - |
| Other | 2 | 1 | 3 |
| **Language (*n*)** | | | |
| English | 4 | 3 | 6 |
| Multi-lingual, including English | 4 | 4 | 1 |

**RES™**: Regenerative Epidermal Suspension; **‡**: Active-control group; *****: median (interquartile range); **M**: male; **F**: female; **Min**: minimum; **Max**: maximum; **PTSS**: post-traumatic stress symptoms.

The burn depth was predominantly superficial partial and mid-dermal partial-thickness for all three groups. The percentage TBSA-B of deep partial-thickness burn depth was <0.1%. A comparison of objectively measured burn depth using laser doppler imaging with clinically assessed burn depth was not possible in this cohort of participants due to the limited field of view; the entire burn injury was not captured within a single image. The same reason applied for three-dimensional, objective burn wound stereophotogrammetry. Scald (81.8%) was the predominant injury mechanism, followed by flame and then radiant heat. The anatomical location of these injuries was mainly in the head/neck, trunk, and upper limb. The distribution of lower limb and foot injuries was uneven, with compar-

atively more burns in both these anatomical locations noted in the silver dressing and Biobrane[®] groups than the RES™/Biobrane[®] group. There were no full thickness burn depth wounds or injuries involving the perineum. Only one participant did not receive appropriate first aid, which was defined as 20 minutes of cool running water within three hours of injury [59,60]. The median time from injury to presentation was 5.5 (IQR: 3.0–12.75) hours and at initial dressing application under general anaesthesia was 12.0 (IQR: 6.0–20.25) hours, Table 2. One participant aged older than eight years was enrolled in each group. To preserve the identity of these participants, demographic and secondary outcome data are not presented within this report.

**Table 2.** Baseline burn wound characteristics.

| Parameter (n) | Silver Dressings ‡ (8) | RES™/Biobrane® (7) | Biobrane® (7) |
|---|---|---|---|
| **TBSA-B (%)** | | | |
| Median (IQR) | 11.5 (7.0–15.75) | 6.00 (5.0–20.0) | 10.0 (8.0–15.0) |
| Min–Max | 5–20 | 5–33 | 7–27 |
| **Burn Depth (n)** | | | |
| Superficial thickness | 2 | 1 | 1 |
| Superficial partial thickness | 8 | 7 | 7 |
| Mid-Dermal partial thickness | 4 | 6 | 4 |
| Deep partial thickness | 1 | 1 | 2 |
| **Injury mechanism (n)** | | | |
| Scald | 7 | 6 | 5 |
| Flame | - | 1 | 2 |
| Radiant Heat | 1 | - | - |
| **Anatomical Location (n)** | | | |
| Head/Neck | 8 | 6 | 6 |
| Trunk | 8 | 7 | 7 |
| Upper limb | 5 | 5 | 7 |
| Hand | 3 | 2 | 2 |
| Lower limb | 1 | 3 | 4 |
| Foot | 8 | 2 | 7 |
| **Appropriate First Aid (n)** | 8 | 6 | 7 |
| **Burn age at initial presentation (hours) *** | 8.50 (2.25–21.00) | 9.00 (4.00–12.00) | 4.00 (3.00–6.00) |
| **Burn age at initial dressing application (hours) *** | 13.00 (6.25–24.75) | 14.00 (10.00–20.00) | 6.00 (4.00–18.00) |

**RES™**: Regenerative Epidermal Suspension; ‡: active control group; *: median (interquartile range); **Min**: minimum; **Max**: maximum.

### 3.3. Primary Outcome: Time to Re-Epithelialisation

The median TTRE was the same for RES™/Biobrane[®], 12 days (IQR: 5.6–18.4), and slower by 2 days for Biobrane[®], 14 days (6.3–21.7), when compared to the silver dressing group, 12 days (IQR: 3.7–20.3). The survival distributions for the three groups were not statistically significant $\chi^2$ (2) = 2.218, $p$ = 0.330, Table 3. Wound re-epithelialisation, which was assessed by a masked panel of burn clinicians and the treating burn surgeon, was found to be moderately positively correlated, r (139) = 0.39) [51].

**Table 3.** Primary and secondary outcomes measured up to the primary endpoint of ≥ 95% burn wound re-epithelialisation.

| Outcome (n) | Silver Dressings ‡ (8) | RES™/Biobrane® (7) | Biobrane® (7) |
|---|---|---|---|
| **Time to Re-epithelialisation (days)** | | | |
| Mean TTRE (SD) | 15.1 (±8.87) | 11.1 (±3.28) | 15.1 (±6.04) |
| Median TTRE (95%CI) | 12 (3.7–20.3) | 12 (5.6–18.4) | 14 (6.3–21.7) |
| **Number of dressing applications (n)** | | | |
| Under general anaesthesia * | 3 (1.00–4.75) | 2 (1.00–3.00) | 3 (2.00–4.00) |
| Awake * | 2 (2.00–4.00) | 4 (3.00–4.00) | 2 (2.00–4.00) |
| Total * | 6 (3.00–7.75) | 6 (5.00–6.00) | 7 (5.00–8.00) |
| **Pain *** | | | |
| *Initial dressing application* | | | |
| Median pre-intervention FLACC score | 2.00 (0.00–2.75) | 0.00 (0.00–3.25) | 0.00 (0.00–0.00) |
| Median post-intervention FLACC score | 0.00 (0.00–0.00) | 0.00 (0.00–4.00) | 0.00 (0.00–2.50) |
| Median difference FLACC score | −2.00 | 0.00 | 0.00 |
| Median pre-intervention NRS-P Proxy score | 5.50 (3.00–9.50) | 7.00 (5.00–10.00) | 4.00 (3.00–4.75) |
| Median post intervention NRS-P Proxy score | 2.00 (1.00–4.00) | 4.00 (0.00–8.50) | 2.50 (0.00–10.00) |
| Median difference NRS-P Proxy score | −3.50 | −3.00 | −1.5 |
| *Dressing Change #1* | | | |
| Median pre-intervention FLACC score | 0.00 (0.00–1.75) | 0.00 (0.00–2.00) | 0.00 (0.00–0.00) |
| Median post-intervention FLACC score | 0.00 (0.00–3.50) | 0.00 (0.00–0.00) | 0.00 (0.00–0.00) |
| Median difference FLACC score | 0.00 | 0.00 | 0.00 |
| Median pre-intervention NRS-P Proxy score | 1.00 (0.00–6.00) | 2.50 (0.00–5.50) | 1.00 (0.50–4.00) |
| Median post intervention NRS-P Proxy score | 4.00 (0.00–5.00) | 2.00 (1.75–3.50) | 1.00 (0.00–2.00) |
| Median difference NRS-P Proxy score | +3.00 | −0.50 | 0.00 |
| **Itch *** | | | |
| Median NRS-I Proxy score | 6.00 (3.00–8.00) | 4.00 (2.25–7.00) | 4.00 (2.00–7.00) |
| **Ease of Dressing Application *** | | | |
| *Initial dressing application* | | | |
| Doctors (n = 48) | | | |
| Application ease | 5.00 (2.00–6.00) | 6.50 (2.75–7.00) | 5.00 (3.00–7.00) |
| Dressing conformability | 3.00 (2.00–7.00) | 3.50 (2.00–7.50) | 4.50 (2.7–6.00) |
| Duration(minutes) | <60 | >60 | >60 |
| Nurses (n = 26) | | | |
| Application ease | 2.00 (1.00–4.00) | 2.50 (1.00–8.75) | 1.00 (0.00–2.00) |
| Dressing conformability | 3.00 (1.00–3.50) | 2.00 (1.00–5.00) | 1.00 (0.50–1.50) |
| Duration(minutes) | <60 | <60 | <10 |
| *Dressing Change* | | | |
| Doctors (n = 43) | | | |
| Application ease | 7.00 (4.25–7.75) | 4.00 (2.00–8.00) | 3.00 (2.00–7.00) |
| Dressing conformability | 7.00 (5.00–7.00) | 2.00 (2.00–7.00) | 2.00 (2.00–7.00) |
| Duration(minutes) | <60 | <60 | 30>, <60 |
| Nurses (n = 101) | | | |
| Application ease | 2.00 (1.00–3.00) | 2.00 (1.00–4.00) | 1.50 (0.00–3.00) |
| Dressing conformability | 2.00 (1.00–3.25) | 2.00 (0.25–4.25) | 2.00 (1.00–3.00) |
| Duration (minutes) | <60 | <60 | 30>, <60 |
| **Intervention Fidelity *** | | | |
| *Initial dressing application (%)* | | | |
| QV wash | 100 | 100 | 100 |
| Intervention | 87.50 | 83.61 | 97.62 |
| *Dressing Change (%)* | | | |
| QV wash | 89.47 | 72.22 | 76.19 |
| Intervention | 94.08 | 90.79 | 100.00 |
| **Treatment Satisfaction *** | | | |
| Staff | 9.00 (8.75–10.00) | 9.00 (8.00–9.75) | 8.00 (2.25–9.00) |
| Parent/Guardian | 10.00 (9.00–10.00) | 10.00 (10.00–10.00) | 10.00 (9.00–10.00) |

RES™: Regenerative Epidermal Suspension; ‡: active control group; *: median (interquartile range); **TTRE**: time to re-epithelialisation; **SD**: standard deviation; **CI**: confidence interval; **NRS-P**: Numeric Rating Scale Pain; **NRS-I**: Numeric Rating Scale Itch.

*3.4. Secondary Outcomes*

3.4.1. Pain

Proxy pain scores were reported for all participants by nurses and parents/guardians using the FLACC and NRS-P Proxy pain scores, respectively. At the initial dressing application, the silver dressing group had the largest reduction in pre- to post-intervention pain (FLACC median difference—2.00; NRS-P Proxy median difference—3.50). However, at the first dressing change, only the RES™/Biobrane® group had a reduction in pain (median difference in NRS-P Proxy—0.50). There was no change in the reported FLACC pain score at this timepoint. The total median number of dressing applications was similar for all three groups (Silver dressings: 6.00 (IQR: 3.00–7.75); RES™/Biobrane®: 6.00 (IQR: 5.00–6.00) and Biobrane®: 7.00 (IQR: 5.00–8.00). The RES™/Biobrane® group had the lowest number of dressing applications under a general anaesthetic, with a median of 2.00 (IQR: 1.00–3.00), and the most dressing applications while awake, with a median of 4.00 (IQR: 3.00–4.00), Table 3.

3.4.2. Itch Intensity

The median itch scores reported by parents/guardians during dressing changes were two points higher (worse) on a 0 to 10 NRS in the silver dressing group (NRS-I Proxy; median 6.00 (IQR: 3.00–8.00)) compared to the other two groups. Self-reports of pain and itch were not analysed due to low responses (only one participant per group of an eligible age to self-report), Table 3.

3.4.3. Ease of Dressing Application

At the initial dressing application, doctors reported that application ease was the most difficult for the RES™/Biobrane® groups (median of 6.50 (IQR: 2.75–7.00)). The dressing conformability was the easiest (median of 3.00 (IQR: 2.00–7.00)) and quickest (<60 min) in the silver dressing group. Nurses reported that at initial dressing application both the application ease (median of 1.00 (IQR: 0.00–2.00)) and conformability (median of 1.00 (IQR: 0.50–1.50)) were the easiest and the fastest (<10 min) for the Biobrane® group. During subsequent dressing changes, doctors reported that the application ease was most difficult (median of 7.00 (IQR: 4.25–7.75)) with the least conformability (median of 7.00 (IQR: 5.00–7.00) in the silver dressing group. The dressing application ease reported by nurses was the easiest in the Biobrane® group (median of 1.50 (IQR: 0.00–3.00)) and had a similar conformability across all groups. The duration of dressing application was the shortest in in the Biobrane® group (between 30–60 min), as reported by both doctors and nurses during dressing changes, Table 3.

From the free text response section in the ease of dressing application questionnaire, 218 responses were collected in total. At baseline, 34% (74/218) of the responses were collected from doctors (48/74) and the remainder from nurses (26/74). At subsequent dressing changes, 66% (144/218), most of the responses, were collected from nurses (101/144), followed by the doctors (43/144). The responses were divided into two groups based on the time of collection: initial dressing application and subsequent dressing changes. At the initial dressing application, no comment was noted, mainly in the silver dressing group (17/25, 68%), followed by the Biobrane® (11/23, 48%) and RES™/Biobrane® (12/26, 46%) groups. The responses were divided into two themes: (1) factors impacting dressing application and (2) procedure related. The categories related to these themes were divided into three groups: (1) dressing-related, (2) patient-related and (3) staff-related. Finally, the individual descriptors were grouped as positive, neutral, or negative. During dressing changes, the percentage of participants who left the free text section blank was 70% in the silver dressing group (33/47, 70%), 68% in the Biobrane® group (41/60) and 57% in the RES™/Biobrane® (21/37) group. While factors that impacted the dressing application differed slightly per group, the anatomical location (non-planar, curved surfaces such as the head, neck, and limbs) was still the most reported factor in all three groups. The TBSA-B of the burn and the lengthy duration of dressing application were factors that impacted the

dressing application in the silver dressing and RES™/Biobrane® groups. This contrasted with the Biobrane® group in which the sedation level and dressing failures were identified as factors that impacted the dressing application.

### 3.4.4. Intervention Fidelity

The silver dressing group had the highest proportion of intervention fidelity to soap-free non-excisional wound debridement (89.74%). The intervention fidelity was highest in the Biobrane® group at the initial dressing application (97.62%) and subsequent dressing changes (100%) when compared to >80.00% at the initial dressing application and >90.00% at dressing changes for the silver dressing and RES™/Biobrane® groups, Table 3.

### 3.4.5. Treatment Satisfaction

At the primary endpoint of ≥95% re-epithelialisation, all parents/guardians were extremely satisfied with the treatment received, Table 3. Doctors and nurses were marginally more satisfied with the silver dressings and RES™/Biobrane® than with Biobrane®, Table 3. Twelve months post injury at the secondary endpoint, Doctors were the least satisfied with Group C (Biobrane®) and were similarly satisfied with the treatment for Group A (Silver dressings) and Group B (RES™/Biobrane®), Table 4. The parents/guardians were extremely satisfied with the treatment received for both Group A (Silver dressings) and Group B (RES™/Biobrane®) and very satisfied with the Group C (Biobrane®) treatment at the 12-months endpoint, Table 5.

### 3.4.6. Scar Severity

The median scar thickness was similar in all three groups at 12-months post injury, Table 3. The median scar colour was closest to normal skin in the RES™/Biobrane® group ($L^*_{scar}$ 35.78 (15.80–41.26) vs. $L^*_{normal}$ 32.18 (16.53–46.78)) and differed the most from the normal skin colour in the Biobrane® group ($L^*_{scar}$ 39.13 (30.05–45.00) vs. $L^*_{normal}$ 43.73 (33.80–46.58)). The $L^*_{normal}$ for each group was also different; this could be explained by the differences in the Fitzpatrick skin type or the amount of ultraviolet light exposure at the site. The three and six month scar thickness and colorimetry data are included in Supplementary Files S1 and S2. The overall opinion of the burn scar severity reported by parents/guardians of children younger than eight years was similar in all three groups at the three, six and 12-month reviews. However, the parents/guardians reported the least severe burn scars in the RES™/Biobrane® group (median POSAS score of 9.00 (IQR: 4.00–13.00)). This contrasts with the much higher severity of scars reported in the Biobrane® group (median POSAS score of 22.50 (IQR: 14.75–30.25)) and the low scar severity reported in the silver dressing group (median POSAS score of 14.00 (IQR: 6.00–27.00)), Table 4. The parent/guardian-reported scar severity data for participants <8 years at three and six months post injury are included in Supplementary Files S3 and S4.

**Table 4.** Long-term secondary outcomes at secondary endpoint, at 12-months post burn injury, as reported by clinicians.

| Outcome | Silver Dressings | | RES™/Biobrane® | | Biobrane® | |
|---|---|---|---|---|---|---|
| | Median (IQR) | Mean (SD) | Median (IQR) | Mean (SD) | Median (IQR) | Mean (SD) |
| **Scar Characteristics** | | | | | | |
| Thickness (*n*, mm) | 5 | | 4 | | 4 | |
| | 1.32 (0.85–2.92) | 1.77 (1.22) | 1.08 (0.97–1.49) | 1.18 (0.29) | 1.59 (1.35–2.94) | 1.96 (0.94) |
| Colour (*n*) | 4 | | 4 | | 5 | |
| $L^*_{Scar}$ | 32.10 (15.61–40.90) | 29.54 (14.06) | 35.78 (15.80–41.26) | 30.95 (14.46) | 39.13 (30.05–45.00) | 38.06 (8.18) |
| $L^*_{Normal}$ | 29.84 (26.60–40.89) | 32.44 (8.12) | 32.18 (16.53–46.78) | 31.53 (16.10) | 43.73 (33.80–46.58) | 41.37 (7.26) |
| $a^*_{Scar}$ | 14.23 (9.91–18.70) | 14.28 (4.62) | 13.35 (10.05–16.84) | 13.41 (3.55) | 15.12 (13.07–17.57) | 15.25 (2.34) |
| $a^*_{Normal}$ | 13.53 (12.57–14.38) | 13.49 (0.98) | 13.24 (11.17–15.76) | 13.24 (2.57) | 11.23 (10.16–15.88) | 12.43 (3.40) |
| $b^*_{Scar}$ | 13.94 (10.88–16.85) | 13.89 (3.09) | 14.29 (8.88–16.87) | 13.35 (4.30) | 9.35 (3.28–18.11) | 10.24 (7.84) |
| $b^*_{Normal}$ | 16.87 (15.02–18.27) | 16.72 (1.73) | 15.61 (10.91–18.26) | 14.93 (3.95) | 14.85 (12.77–18.94) | 15.52 (3.33) |

**Table 4.** *Cont.*

| Outcome | Silver Dressings | | RES™/Biobrane® | | Biobrane® | |
|---|---|---|---|---|---|---|
| | **Median (IQR)** | **Mean (SD)** | **Median (IQR)** | **Mean (SD)** | **Median (IQR)** | **Mean (SD)** |
| **Clinician Scar Severity Report** OSAS(*n*) ‡ | 5 | | 4 | | 4 | |
| Overall Opinion | 2.00 (2.00–5.50) | 3.40 (3.13) | 2.00 (1.25–2.75) | 2.00 (0.82) | 2.50 (2.00–6.75) | 3.75 (2.87) |
| Treatment Satisfaction | | | | | | |
| Doctor (*n* ¥) | 7 | | 9 | | 7 | |
| | 8.00 (7.00–9.00) | 7.57 (2.23) | 9.00 (8.50–9.50) | 9.00 (0.71) | 5.00 (4.00–15.00) | 5.57 (1.90) |
| Nurse (*n* ¥) | 0 | | 1 | | 0 | |
| | N/A | | 9.00(9.00–9.00) | | N/A | |
| Occupational Therapist (*n* ¥) | 1 | | 1 | | 3 | |
| | 8.00 (8.00–8.00) | 8.00 (_) | 7.00 (7.00–7.00) | 7.00 (_) | 8.00 (8.00–_) | 8.33 (0.58) |

**RES™**: Regenerative Epidermal Suspension; **IQR**: interquartile range; **SD**: standard deviation; **mm**: millimetre; **L\***: lightness; **a\***: erythema; **b\***: pigmentation; **OSAS**: Observer Scar Assessment Scale; *n*: number; **N/A**: not applicable; ‡: OSAS completed for participants <8 years old; ¥: ≥1 clinician completed a treatment satisfaction rating per participant.

### 3.4.7. Health-Related Quality of Life

The parents/guardians reported little to no impact of the burn injury on the health-related quality of life in children <8 years who were assigned silver dressings or RES™/Biobrane® at 12-months post injury, Table 5. Similarly, parents/guardians reported that the physical symptoms had some impact on the HRQoL in children allocated to Biobrane® at one year post injury, Table 5. Most of the impacts of HRQoL were noted at three months post injury by the parent/guardian for all three groups, Supplementary File S5. Scar sensitivity had the most impact for children in the Biobrane® group at three months post injury. When compared to the silver dressing or Biobrane® groups, parents/guardians reported the least impact of the burn injury on the scar-specific HRQoL in children <8 years that were assigned to RES™/Biobrane® at three, six and 12-months post injury, Supplementary File S5. At the three month follow-up, the median generic paediatric HRQoL using the CHU-9D was similar for participants in the RES™/Biobrane® group, with a score of 1.00 (IQR: 0.77–1.00), and the silver dressing group, with a score of 0.94 (IQR: 0.76–1.00). It was the lowest in the Biobrane® group, with a score of 0.75 (IQR: 0.44–0.93). The paediatric HRQoL reported by the parents/guardians for participants younger than eight years using the CHU9D was similar in all three groups at the six and 12-month follow-up (Supplementary File S6).

**Table 5.** Long-term secondary outcomes at secondary end point, 12-months post burn injury, as reported by a parent/guardian.

| Outcome | Silver Dressings | | RES™/Biobrane® | | Biobrane® | |
|---|---|---|---|---|---|---|
| | **Median (IQR)** | **Mean (SD)** | **Median (IQR)** | **Mean (SD)** | **Median (IQR)** | **Mean (SD)** |
| POSAS † (*n*) | 7 | | 5 | | 4 | |
| Overall Opinion | 3.00 (1.00–6.00) | 3.86 (2.97) | 2.00 (1.00–3.50) | 2.20 (1.64) | 3.50 (3.00–5.50) | 4.00(1.41) |
| POSAS Score | 14.00 (6.00–27.00) | 18.00 (14.06) | 9.00 (4.00–13.00) | 8.60 (5.18) | 22.50 (14.75–30.25) | 22.50(8.27) |
| BBSIP † (*n*) | 7 | | 5 | | 4 | |
| Overall impact of burns | 1.13 (1.00–3.00) | 1.71 (0.95) | 1.00 (1.00–1.00) | 1.00 (0.00) | 1.38 (1.09–1.56) | 1.34 (0.26) |
| Treatment Satisfaction ≠ (*n*) | 10 | | 8 | | 6 | |
| Parent/Guardian | 10.00 (8.25–10.00) | 9.10 (1.66) | 10.00 (10.00–10.00) | 10.00 (0.00) | 9.50 (9.00–10.00) | 9.50 (0.55) |

**RES™**: Regenerative Epidermal Suspension; **IQR**: interquartile range; **SD**: standard deviation; **POSAS**: Patient Observer Scar Assessment Scale; *n*: = number; †: POSAS completed for participants < 8 years old; ≠: ≥1 parent/guardian completed a treatment satisfaction rating per participant.

### 3.4.8. Health Resource Utilisation

At the initial dressing application, participants in the RES™ /Biobrane® group incurred the highest median intervention cost at AUD 9262.00 (IQR: AUD 8830.10–AUD 21,280.70), followed by AUD 667.92 (IQR: AUD 498.97–AUD 1361.72) in the Biobrane® group and lastly, AUD 291.30 (IQR: AUD 194.02–AUD 438.05) for the silver dressing group. This contrasts with the median intervention cost during dressing changes of AUD 645.87 (IQR: AUD 278.70–AUD 933.43) in the silver dressing group, followed by AUD 385.38 (IQR: AUD 261.33–AUD 666.81) in the Biobrane® group and the lowest cost for participants in the RES™ /Biobrane® group of AUD 63.91 (AUD 28.34–AUD 891.83), see Supplementary File S7. Over the 12-month period of participation, the median hospitalisation cost in the outpatient department was similar in all three groups: AUD 314.16 (IQR: AUD 304.87–AUD 444.67) for silver dressings, AUD 314.57 (IQR: AUD 304.87–AUD 424.34) for the RES™/Biobrane® group and AUD 314.16 (IQR: AUD 304.87–AUD 380.71) for the Biobrane® group. The median hospitalisation cost in the emergency department and inpatient admission were the lowest in the RES™/Biobrane® group at AUD 642.29(IQR: AUD 367.99–AUD 1270.38) and AUD 21,707.34 (IQR: AUD 4424.21–AUD 76,438.30), respectively, when compared to the silver dressing and Biobrane® groups (Supplementary File S7).

### 3.4.9. Adverse Events

Overall, 73% (*n* = 16) of participants experienced adverse events of varied severity. The silver dressing group had *n* = 10 adverse events, RES™/Biobrane® group had *n* = 6 adverse events and the Biobrane® group had *n* = 9 adverse events. One wound infection, one unplanned ICU admission and no requirements for skin graft were reported in the RES™/Biobrane® group. Most of the adverse events were protocol deviations in the RES™/Biobrane® and Biobrane® groups in which the Biobrane® was partially non-adherent and was thus replaced with silver dressings. In contrast, there were three wound infections, two unplanned ICU admissions and one skin graft required in the Biobrane® group. In the silver dressing group, there were two wound infections, one case of wound sepsis, two unplanned ICU admissions and two skin grafts required, Table 6. Other adverse events were: 12-hour delay to operation theatre (*n* = 1, Silver dressings group), fever associated with teething (*n* = 1, RES™/Biobrane® group) and unplanned admission (n = 1, RES™/Biobrane® group), Table 6. The severity of adverse events was graded using the Clavein–Dindo scale post-hoc. This demonstrated that the more severe complications were noted in participants allocated to silver dressings (*n*= 4) and Biobrane® (*n* = 4) than the RES™ /Biobrane® (*n* = 1).

**Table 6.** Adverse events.

| Type of Adverse Event | Silver Dressings ‡ | RES™/Biobrane® | Biobrane® |
|---|---|---|---|
| (*n*) | (13) | (8) | (10) |
| Nil | 3 | 2 | 1 |
| Wound infection | 2 | 1 | 3 |
| Allergic reaction | 1 | 0 | 0 |
| Sepsis | 1 | 0 | 0 |
| Unplanned ICU admission | 2 | 1 | 2 |
| Burn depth progression | 2 | 1 | 3 |
| Required split-thickness skin graft | 2 | 0 | 1 |
| Other | | | |
|    Fever associated with teething | 0 | 1 | 0 |
|    Unplanned ward admission | 0 | 1 | 0 |
|    12-h delay to theatre | 1 | 0 | 0 |

**Table 6.** *Cont.*

| Clavein–Dindo Grade of Complication | Silver Dressings ‡ | RES™/Biobrane® | Biobrane® |
|---|---|---|---|
| (*n*) | (10) | (6) | (9) |
| Grade I | 2 | 5 | 1 |
| Grade II | 0 | 0 | 2 |
| Grade IIIa | 0 | 0 | 0 |
| Grade IIIb | 4 | 0 | 3 |
| Grade IVa | 0 | 0 | 0 |
| Grade IVb | 4 | 1 | 3 |
| Grade V | 0 | 0 | 0 |
| **Clavein–Dindo Grade of Most Severe Complication** | **Silver Dressings ‡** | **RES™/Biobrane®** | **Biobrane®** |
| (*n*) | (5) | (5) | (6) |
| Grade I | 1 | 4 | 1 |
| Grade II | 0 | 0 | 2 |
| Grade IIIa | 0 | 0 | 0 |
| Grade IIIb | 2 | 0 | 1 |
| Grade IVa | 0 | 0 | 0 |
| Grade IVb | 2 | 1 | 2 |
| Grade V | 0 | 0 | 0 |

RES™: Regenerative Epidermal Suspension: ‡: active control group; **ICU**: intensive care unit.

## 4. Discussion

The findings indicated that burn wounds achieved ≥95% re-epithelialisation in a median of two days slower in wounds treated with Biobrane® only when compared to a median TTRE for burn wounds treated with RES™/ Biobrane® and silver dressings, which demonstrated the same TTRE. The protocol pre-specified that four days would be considered a clinically meaningful difference [14]. Thus, based on the findings from 22 participants, a clinically meaningful difference in the burn wound TTRE was not demonstrated. A reduction in pre- and post-intervention median pain scores was reported in all three intervention groups at the initial dressing application. At the first dressing change, only the parents/guardians of the children whose burn wounds were treated with RES™/Biobrane® reported a reduction in the pre- to post-intervention median pain scores. For participants <8 years assigned RES™/Biobrane®, the parents/guardians reported the least impact of the burn injury on the scar-specific HRQoL at all three long-term follow-up visits when compared to the participants allocated silver dressings or Biobrane®. The intervention cost for participants in the RES™/Biobrane® groups was the highest at the initial dressing application and the lowest during dressing changes in comparison to the silver dressing and Biobrane® groups. While the hospitalisation cost was similar for all three groups during outpatient visits, participants allocated to the RES™/Biobrane® group incurred the lowest hospitalisation cost during emergency department and inpatient admissions when compared to the silver dressing and Biobrane® groups. When compared to the silver dressing or Biobrane® (Clavein–Dindo Grade I, *n* = 1) groups, more participants sustained less severe adverse effects when assigned to RES™/Biobrane® (Clavein–Dindo Grade I, *n* = 4), Table 5. The higher number of wound infections (*n* = 3), burn depth progression (*n* = 3), darker skin type and slower time to initial presentation may explain the slower TTRE for burn wounds in the Biobrane® group as these are all risk factors for delayed re-epithelialisation.

All initial dressing applications were completed under general anaesthesia with perioperative prophylactic antibiotic administration. This facilitated a complete, non-excisional debridement of all non-viable skin from the burn wound, ideal pain control and an optimal application of the assigned intervention to the burn wound. At the first dressing change, the marked increase (+3.00) in reported pre- and post-intervention pain scores for the silver dressing group may be attributed to the complete removal and replacement of the dressings

for children in the Biobrane® group. This contrasted with the change of the outer secondary dressings and trimming of the Biobrane® in the intervention arms, hence the reduction in pain scores of 0.50 in the RES™/Biobrane® group. Keratinocytes exert important immune functions during re-epithelialisation via cell signalling between keratinocytes and immune cells, direct interaction with T-cells through antigen presentation and the production of anti-microbial peptides [61]. It is possible that the autologous keratinocytes contained in RES™ [62], which was applied to burn wounds in the RES™/Biobrane® group, was a protective factor that allowed for the fewest infections in this intervention group when compared to the silver dressing and Biobrane® groups. Alternatively, differences in baseline demographics (particularly burn location and co-morbidities), which were unable to be controlled for due to the small sample size, may have accounted for the differences between the groups as opposed to the interventions.

The responses obtained from doctors and nurses regarding the ease of dressing application added to the understanding of the clinical context of the evaluated interventions. '*Factors that impact dressing application*' was the most common theme, with '*anatomical location*', '*application ease*', and '*conformity*' being the most common descriptors. As most of the burn injuries were sustained in toddlers (median age of 1–2 years) due to scald injuries (*n* = 18) distributed mainly in the upper torso (head/neck *n* = 20; trunk *n* = 22; upper limb *n* = 17; hand *n* = 7), it is not surprising that the anatomical location was identified as impacting the dressing application, Tables 1 and 2. The head, neck and hands are often challenging areas to achieve acceptable conformity and application ease during the dressing of small children. Infection and partial dressing failure, resulting in protocol deviation, were specifically referred to in the Biobrane® group at dressing changes. Off-protocol management was a descriptor that was also noted in RES™/Biobrane® group. This occurred when there was a partial or complete failure of the Biobrane® to adhere to the burn wound surface, resulting in a change of the dressing to silver dressings (active control group). This usually occurred at the first or second dressing change within the first week of injury. It is important to note that these descriptors were deduced from a small sample of responses. Further evaluation is required to understand whether these descriptors were a result of a varied clinician familiarity with the intervention, the anatomical location where the dressing was applied, or other factors that were not mentioned.

To date, apart from the BRACS trial, the trials evaluating RES™ for the management of paediatric burn injuries are two in Australia [17,63] and one in the United States of America [64]. One has completed data collection [17], and two are still in data collection [63,64]. Following randomisation, participants that were assigned standard treatment in the pilot three-armed trial conducted by Wood et al. [17] were not taken to the operating theatre for the application of silver-impregnated dressings compared to Biobrane® and RES™/Biobrane® groups, which were taken to the operating theatre for the intervention. In contrast to the trial by Wood et al. [17], the initial dressing application for all participants enrolled in the BRACS trial, irrespective of allocated intervention, occurred under general anaesthesia in the operating theatre. Thus, all participants were given the same initial wound management, which may have been why the median wound TTRE for each intervention group was within two days of the overall median TTRE for the study cohort. The trial based in the United States of America, currently in recruitment, is a two-arm, parallel-group, multicentre randomised trial investigating the safety and effectiveness of RES™ compared to the standard of care treatment for partial-thickness burn injuries in children aged one to sixteen years old [64]. The primary outcome for this study is an incidence of ≥95% re-epithelialisation at day ten and day twenty-eight. The secondary outcomes include the incidence of a conventional skin graft, burn pain, burn area, burn itch, scar severity, investigator treatment preference and health resource utilisation. The trial based in Australia, which is still in recruitment, is a two-armed, parallel-group, randomised controlled pilot trial evaluating the pigmentation of split-thickness skin graft donor sites treated with RES™ at 12-months post skin graft in children [63]. The primary outcome is the donor site pigmentation at 12-months post skin graft, with secondary outcomes including

the time in days to ≥95% re-epithelialisation, pain, itch, intervention fidelity, treatment satisfaction scar severity, health-related quality of life and health resource utilisation. This is the only trial evaluating paediatric split-thickness skin graft donor sites treated with RES™.

There were several limitations in this study. The recruitment was suspended indefinitely before meeting the target sample size due to the COVID-19 pandemic, resulting in a small cohort sample, and limiting the data analysis to a descriptive synthesis of findings. Thus, the results should be interpreted with notable caution and as a pilot study that provides direction for a future full randomised controlled trial. Another temporary change in practice, brought about at the study site during the COVID-19 pandemic, was that the spray application of RES™ was not permitted. Application was thus changed to a droplet application method. RES™ is marketed for both droplet and spray application. This change in application resulted in more likelihood of the run-off of RES™ from curved burn wound surfaces, such as limbs, as opposed to the better distribution of RES™ when it is applied as spray. Consequently, the Biobrane® was partially applied at the dependent wound edge for the faster retention of the RES™, especially on a curved burn wound surface. In addition, policy changes at the study site as a result of the COVID-19 pandemic also affected the long-term data collection as telehealth appointments were increased during this period, resulting in the loss of objective data collection or a loss of follow-up. However, these factors, which had the potential to influence the outcomes, should have been balanced across the groups due to the randomisation.

The recruitment rate was low at 41%, with 28 potentially eligible participants not being included in the cohort for the following reasons: the burn surgeon declined or did not consider the participant eligible for enrolment, a non-burn surgeon was on call and delayed referral. The main challenge for these missed cases was that the burn surgeon declined participation as they did not agree that the initial burn wound debridement should be performed under general anaesthesia. In lieu of an initial non-excisional debridement under general anaesthesia, Ketamine-based procedural sedation and analgesia in the emergency department was a common alternative based on a retrospective cohort at the study centre [65]. Accessibility to the operating theatres, especially after hours, was not always guaranteed, and was thus viewed by the burn surgeons as disruptive to the intended wound management plan due to the potential delay of initial debridement and initial definitive wound cover while waiting for a theatre.

Eighty-two percent (*n* = 18/22) of the enrolled cohort completed the study at the 12-month follow-up. The reasons for non-completion at the final review were reported to be due to socio-economic constraints, travel restrictions imposed by the COVID-19 pandemic and a lack of concern from parents/guardians about the burn scar, thus requesting cessation of participation in the trial. Bias as a result of loss to follow could be explained by the understanding that only those children with burn injuries that met the inclusion criteria and who had an accessible parent/guardian that was suitably present and engaged in the trial have any data represented. Since attrition was less than 20% in a very small cohort (*n* = 22) and in the context of burn research, this is not of concern and should be taken into consideration when developing future trials in this field.

Despite the use of previously validated tools, accurate measurement of the burn wound size with three-dimensional camera systems (Intel® RealSense™ D415/Wound Measure and LifeVizII®/DermaPix®) and burn depth measurement using the laser Doppler imager were not possible due the limited field view, which did not capture the entire large burn wound in a single image. A prospective cohort study of 13 children with 25 burn injuries identified that burn wound stereophotogrammetry using the Intel® RealSense™ D415/Wound Measure system was most the challenging when measuring the size of highly contoured burn wounds [66]. Finally, the health resource utilisation required to implement any of the three wound management approaches evaluated in this study can be considered a limitation with applicability to potentially resource-rich centres only. However, the findings from this study provide evidence for consideration at any centre seeking to expand their paediatric burn wound care capacity.

In a recent systematic review and meta-analysis of the efficacy of ASCS in the management of partial-thickness burn injuries and split-thickness skin graft donor sites [15,67,68], only one of the five randomised controlled trials included was completed in children [17]. The meta-analysis reported that in children, ASCS may reduce partial-thickness burn wound re-epithelialisation time. However, the certainty of evidence was low [67]. The systematic review also identified that paediatric burn wounds treated with ASCS had increased odds for infection as well as markedly lower odds for further surgery to manage the burn wound [15]. Participants in the paediatric study had a wait time of two to five hours following the initial presentation at the study centre prior to the initial dressing application in the operating theatre [17]. The impact of initial non-excisional debridement under general anaesthesia of medium-to-large TBSA-B injuries on burn wound re-epithelialisation should be evaluated in a larger cohort of children to understand the applicability of this wound management approach. Inevitably, this will require the evaluation of accessibility of the operating theatre setting, amongst other logistic requirements, for this approach to be implemented in standard practice.

Overall, the BRACS trial results indicate potential benefits of RES™/Biobrane® for the management of partial-thickness paediatric burns of 5% ≥ TBSA-B extending up to mid-dermal partial-thickness burn depth in children under 16 years of age who are treated within 48 hours of injury. Based on the current pilot study, these potential benefits seem to outweigh the risks or burden that could be associated with this wound management approach. As such, the design of a future fully powered randomised controlled trial is warranted and should incorporate the following suggested improvements. Anatomical location and co-morbidities should be controlled for, especially when between-group imbalance occurs, as these factors can impact re-epithelialisation and scar formation from a pathophysiological perspective. Burn size stereophotogrammetry and burn depth measurement by laser doppler imaging are not feasible for medium-to-large burn injuries and thus alternate objective measures should be considered. Accessibility to an operating theatre irrespective of the time of day is an important logistic requirement that should be considered when establishing the setting for the future trial. Masking the assessors of both re-epithelialisation and scar endpoints allows for less potential bias associated with important outcome assessments. Measuring the need for scar rehabilitation as an outcome would enable clinicians to understand the potential health resource utilisation that may be required once burn wound re-epithelialisation is achieved and thus plan accordingly.

## 5. Conclusions

In this small cohort of children with medium- to large-size partial-thickness burns, treatment with RES™/Biobrane® had a re-epithelialisation time that was no different to the control group (silver dressing) and faster by two days when compared to the Biobrane® group. Additionally, burn wounds treated with RES™/Biobrane® had fewer adverse events, the least impact on the scar-specific health-related quality of life as reported by parent/guardians, the highest intervention cost at initial dressing application and the lowest health resource utilisation during inpatient and emergency department hospitalisation. The between-group differences for the reduced TTRE and pre- and post-intervention pain scores, noted in the RES™/Biobrane® group, were neither statistically nor clinically significant. In combination, these results contribute to the existing and emerging body of evidence pertaining to the use of ASCS for management of paediatric thermal injuries. However, a fully powered randomised controlled trial is warranted to obtain a better understanding of these findings.

**Supplementary Materials:** The following supporting information can be downloaded at: https://www.mdpi.com/article/10.3390/ebj4010012/s1. Supplementary File S1: Scar thickness for participants <8 years; Supplementary File S2: Scar colorimetry for participants <8 years; Supplementary File S3: Clinician report of scar severity with OSAS for participants < 8 years; Supplementary File S4: Parent/Guardian report of scar severity with POSAS for participants <8 years; Supplementary File S5: Parent/Guardian report of scar-specific health-related quality of life with BBSIP for participants < 8 years old; Supplementary File S6: Paediatric health-related quality of life based on parent reported CHU9D; Supplementary File S7: health resource utilisation.

**Author Contributions:** All authors attest that they meet the current ICMJE criteria for authorship. Conceptualisation, R.K. and B.G.; data curation, A.B.; formal analysis, A.B., D.V. and S.M.M.; funding acquisition, R.K. and B.G.; investigation, A.B. and R.K.; methodology, A.B., Z.T., R.K., D.V., S.M.M. and B.G.; project administration, B.G.; resources, R.K.; software, A.B.; supervision, Z.T., R.K., S.M.M. and B.G.; writing—original draft, A.B.; writing—review & editing, A.B., Z.T., R.K., D.V., S.M.M. and B.G. All authors have read and agreed to the published version of the manuscript.

**Funding:** This study is supported by a research grant from AVITA Medical Pty Limited (Grant Reference: 2018000489) that is administered by Queensland University of Technology.

**Institutional Review Board Statement:** The study was conducted in accordance with the Declaration of Helsinki and approved by the Institutional Review Boards of Children's Health Queensland Hospital and Health Service Human Research Ethics Committee (HREC/17/QRCH/278, SSA/18/QRCH/41 on 21 December 2017) and Queensland University of Technology (QUT HREC 1800000372 on 30 April 2018) for studies involving humans.

**Informed Consent Statement:** Informed consent was obtained from all subjects involved in the study.

**Data Availability Statement:** Not applicable.

**Acknowledgments:** The authors would like to acknowledge the Pegg Leditschke Children's Burn Centre staff and the enrolled children and their guardians for their participation in this study.

**Conflicts of Interest:** A.B. conducted this trial as part of a Ph.D. degree and has no financial interest in the products used in this trial. A co-investigator (B.G.) was partially funded through the QUT-administered AVITA Medical Ltd. grant. AVITA Medical was not involved in the study design, conduct, analysis or reporting of this trial.

**Registration:** The trial was prospectively registered with the Australian New Zealand Clinical Trials Registry (ACTRN12618000245291) on 15 February 2018 [69].

**Protocol:** The protocol was also published online on 31 October 2019 [14].

## Abbreviations

| | |
|---|---|
| **ACHD** | Autologous Cell Harvesting Device |
| **ASCS** | Autologous Skin Cell Suspension |
| **AUD** | Australian Dollar |
| **BBSIP** | Brisbane Burn Scar Impact Profile |
| **BRACS** | Biobrane®, RECELL® Autologous Skin Cell Suspension and Silver Dressings |
| **CHU9D** | Child Health Utility 9D |
| **COD** | Change of Dressing |
| **FLACC** | Face, Legs, Activity, Cry, Consolability |
| **FPS-R** | Faces Pain Scale-Revised |
| **HREC** | Human Research Ethics Committee |
| **IQR** | Interquartile Range |
| **MAX** | Maximum |
| **MIN** | Minimum |
| **NRS-I** | Numeric Rating Scale—Itch |
| **NRS-I Proxy** | Numeric Rating Scale—Itch Proxy |
| **NRS-P** | Numeric Rating Scale—Pain |
| **NRS-P Proxy** | Numeric Rating Scale—Pain Proxy |

| OSAS | Observer Scar Assessment Scale |
|---|---|
| **POSAS** | Patient and Observer Scar Assessment Scale |
| **QUT** | Queensland University of Technology |
| **REDCap** | Research Electronic Data Capture |
| **RES™** | Regenerative Epidermal Suspension |
| **SSA** | Site Specific Approval |
| **TBSA-B** | Burn Total Body Surface Area |
| **TTRE** | Time to re-epithelialisation |

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
