# Peer review of "A Pilot Randomised Controlled Trial Evaluating a Regenerative Epithelial Suspension for Medium-Size Partial-Thickness Burns in Children: The BRACS Trial"

_2673-1991, doi:10.3390/ebj4010012_

Round 1

Reviewer 1 Report

This manuscript is a preliminary report of an ambitious study, where silver dressing, Biobrane and Biobrane with Recell cell suspension were compared on children 16 years or younger.

The study is well planned, and the recruitment of the patients started 5/2018 but was suspended indefinitely 3/2020 due to Covid pandemic.

This, and also the reasons very well explained, resulted that only 22 patients were recruited in the final study group of three arms. The biggest problem is that the power of the study suffered essentially due to these unfortunate circumstances.

The question is are the authors going to continue the study? If yes, for me this is very extensive report for a pilot study and could be shorter. The other thing is that if this will be the final report this form is more understandable. In any case I have some question that I liked to be answered:

-The protocol allowed the surgeons to decide how often they did the dressing changes in silver dressing group. If the dressings were changed very often, could this affect the epithelialization or even be harmful by removing the newly formed thin epithel?

-The 95% epithelialization was judged by one surgeon and also by a panel of experts on pictures? Judging epithelialization from pictures is very demanding, how succesfull was the 3D-measuring?

-There were also patients that needed skin grafting and the epithelialization was assessed to be 28d. The decision was made by the surgeon. Shouldn´t these patients be discarded from the study?

-I could not find the references 22-23.

This is very important and well-planned study. With so few patients and so many confounding factors the conclusions lack power. I really liked to see this study fulfilled but as such it is now, I recommend making a shorter version, more like a pilot study.

Reviewer 2 Report

I think that the topic is of interest and worth to be published; I have only one minor issue: the limitations are not really mentioned and discussed in a profound way. Please do so! What is the real clinical impact? Please state and discuss.

Reviewer 3 Report

The topic is of interest and importance to the medical community, and I applaud you for this attempt. This is an interesting and important article. It is well written and has a nice touch as it takes a very problematic part of our task (treating pediatric burns) and clarifies it. Having said that, the article does have some flaws that to my understanding could be attended to and that can make it clearer and more precise.

The abstract is not clear enough and does not let the reader understand the main benefits and limitations of the study. It is mainly seen in the fact that from the abstract one cannot understand how limited the study and its' results is.

The research hypothesis and research objectives are detailed at the end of the introduction and make it easy for the reader to understand where the article is heading, but the introduction is not easily read, it drifts between issues and does not put enough importance upon the growing importance of what are the best explored ways to treat pediatric burns. The bibliography is quite extensive, but not up to date, and could be enhanced using the literature regarding burns in adults.

The protocol and research tools are well defined and are clear, understandable, and easy to reproduce, but it is unclear how did you change the procedures due to COVID.

Looking at the study population it seems odd they have not mentioned its extent prior to the COVID-19. Please explain how many patients were seen in your clinic and hospital in 2019 for example.

There is a nice calculation of the sample size/power of the study prior to COVID, but why didn't you try the other way once things have changed due to COVID – that is please do a power calculation to the final sample size. This is even more noticeable when looking at Table 2 where there is a no significance to any of the mentioned factors. Is this is due solely to the size of a small sample? Another related issue here is a selection bias due to the fact those who were lost to follow-up might be less content with their results.

The study protocol is referred to as a bibliographic item, I think it would be better to add it as an appendix and thus aid the reader look up specific issues (like type of prophylactic AB used, the exclusion criteria, how did they see if those not enlisted were similar to all the patients who were included in the study regarding basic demographics etc.)

Another problem is the issue of complications, which is very detailed but does not address any statistical comparison. This could be done according to the Clavien-Dindo or similar scales.

Discussion - A bit long, and you discuss what suits them and less about issues and studies that do not, such as the fact that they are unable to show any real statistical significance between the groups. This is probably due to the unsatisfactory sample size.

You state that this is a pilot study on the subject, and that is not the issue. You lots of interesting things learned from this study which be stated, as well as humbly acknowledging the fact that the very small sample size does not allow them anything but preliminary results and understanding what needs to be done to make this very important study meaningful.

In the limitations, you do discuss the size of the sample, but you say that statistics are enough to correct this, and this is not the case - there is a selection bias that needs to be addressed in the study design and this was not the case.

Another limitation is the fact this is a single institution in a very specific part of the world which makes the generalizability of their results more problematic.

Another issue is the fact they talk about long-term results, but your follow-up period (on less than half of their patients) is no more than 1 year. At best this is mid-term results. 

Round 2

Reviewer 1 Report

There are some important changes done and as such it is ok, although I still think that the paper is far too long but I leave it to the editor to make the final decision.